# ReFusion: Improving Natural Language Understanding with Computation-Efficient Retrieval Representation Fusion

**Shangyu Wu**[1]*, **Ying Xiong**[2]*, **Yufei Cui**[3]*†,
**Xue Liu**[3], **Buzhou Tang**[2], **Tei-Wei Kuo**[45], **Chun Jason Xue**[5]
[1] City University of Hong Kong    [2] Harbin Institute of Technology, Shenzhen
[3] MILA, McGill University    [4] National Taiwan University
[5] Mohamed bin Zayed University of Artificial Intelligence

## Abstract

Retrieval-based augmentations (RA) incorporating knowledge from an external database into language models have greatly succeeded in various knowledge-intensive (KI) tasks. However, integrating retrievals in non-knowledge-intensive (NKI) tasks is still challenging. Existing works focus on concatenating retrievals with inputs to improve model performance. Unfortunately, the use of retrieval concatenation-based augmentations causes an increase in the input length, substantially raising the computational demands of attention mechanisms. This paper proposes a new paradigm of RA named **ReFusion**, a computation-efficient **Re**trieval representation **Fusion** with bi-level optimization. Unlike previous works, ReFusion directly fuses the retrieval representations into the hidden states of models. Specifically, ReFusion leverages an adaptive retrieval integrator to seek the optimal combination of the proposed ranking schemes across different model layers. Experimental results demonstrate that the proposed ReFusion can achieve superior and robust performance in various NKI tasks.

## 1 Introduction

Recent advances in language models (Khandelwal et al., 2020; Borgeaud et al., 2022b; Guu et al., 2020; Lewis et al., 2020; Li et al., 2022) have demonstrated that retrieval-based augmentations (RA) can achieve remarkable performance on a variety of knowledge-intensive (KI) tasks. The basic idea of RA is first leveraging approximate-nearest-neighbor-search-based indexing to retrieve the top-$k$ related knowledge from an external key-value store database, then incorporating the retrieved knowledge into language models with different fusion methods. KI tasks such as question-answering and text generation have an inherent retrieval-based property (Chen et al., 2017; Karpukhin et al., 2020) as answers can be sourced or deduced from external knowledge databases.

However, RA in non-knowledge-intensive (NKI) tasks, such as text classification, are still challenging. Unlike KI tasks, NKI tasks often require understanding and categorizing given only one sentence rather than generating answers with contexts (Wang et al., 2019). Previous works (Guo et al., 2023b; Izacard & Grave, 2021) have proposed to leverage plain texts such as Wikipedia to build the retrieval database. These methods concatenate retrievals with inputs as the context for the model to make predictions. However, these methods significantly increase the input length, imposing considerable computational overheads in the attention module. Besides, the constraints on the model's max input length would also limit the concatenation of retrievals, which may result in a truncated context and lead to semantically incomplete information, thus degrading performance.

This paper introduces a new paradigm of retrieval-based augmentation named **ReFusion**, a computation-efficient **Re**trieval representation **Fusion** framework with bi-level optimization. Different from previous retrieval-based augmentations (Borgeaud et al., 2022b; Guo et al., 2023b),

---

*Authors contributed equally to this research

†Corresponding author

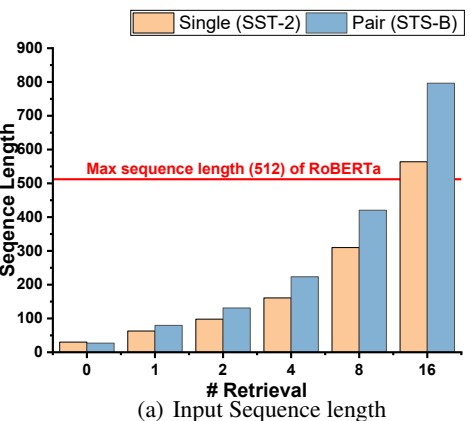 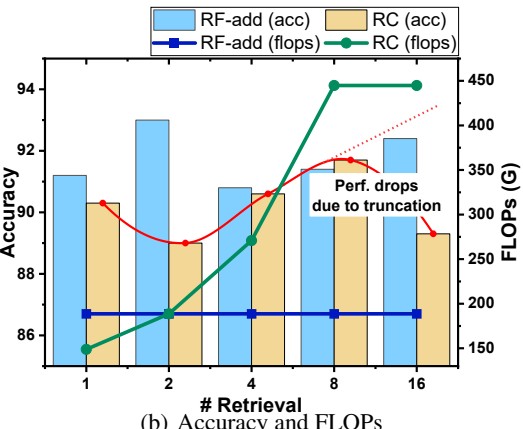

(a) Input Sequence length  (b) Accuracy and FLOPs

Figure 1: Impact of the number of concatenated retrievals on inputs and its effect on model's accuracy and FLOPs. RC (acc) and RF-add (acc) refer to the accuracy of retrieval-concatenation-based augmentation (RC) and retrieval representation fusion with addition (RF-add). RC (flops) and RF-add (flops) refer to the FLOPs of RC and RF-add.

ReFusion directly fuses the representations of retrievals into the hidden states of models. ReFusion consists of three types of modules: the retriever module, the retrieval fusion module, and the adaptive retrieval integrator. Specifically, the retriever module includes a query encoder to encode query texts and an efficient retriever for retrieving the representations of similar information. The retrieval fusion module includes two effective ranking schemes, the reranker-based scheme and the ordered-mask-based scheme, to refine the representation of retrievals. The refined representations are subsequently fused into the hidden states. The adaptive retrieval integrator aims to find the optimal fusion structure with bi-level optimization.

Finally, we conducted comprehensive experiments on 15 different NKI tasks. These tasks vary from sentiment analysis, opinion polarity, natural language inference, etc. The experimental setting follows Gao et al. (Gao et al., 2021). Experimental results show that the ReFusion outperforms other comparisons and achieves superior results on various tasks. Codes are available at [1].

The main contributions of this paper are:

- This paper demonstrates the bottleneck of existing retrieval concatenation-based augmentations, i.e., significant computational overheads and limited performance improvements.
- This paper proposes a new paradigm of retrieval-based augmentation that fuses the retrieval representations into the hidden states of models, making a better trade-off between performance and efficiency.
- This paper designs two ranking schemes for retrieval fusion and an adaptive retrieval integrator for searching the best combination of different ranking schemes.
- Experimental results demonstrate that our ReFusion framework can significantly improve models' understanding capability, achieving superior and robust performance.

## 2 BACKGROUND AND MOTIVATION

**Retrieval-augmented Prompt-based Fine-tuning** The common NKI tasks involves inputting a sentence $x$ as $X_{\texttt{single}} = \texttt{[CLS]}x\texttt{[SEP]}$ or two sentences $x_1, x_2$ as $X_{\texttt{pair}} = \texttt{[CLS]}x_1\texttt{[SEP]}x_2\texttt{[SEP]}$, and outputting a categorical label $y$, where $\texttt{[CLS]}$ is the classification token and $\texttt{[SEP]}$ is sentence separate token in language models. In traditional methods, the $\texttt{[CLS]}$ token is utilized to represent the overall contextual information of the input and to facilitate classification tasks, i.e., $y_{\texttt{logits}} = \texttt{softmax}(W_o \cdot h_{\texttt{[CLS]}})$, where $h_{\texttt{[CLS]}}$ is the final hidden states of $\texttt{[CLS]}$ token. To harness the potential of transformer-based models in terms of mask-prediction capabilities, which serve as the pre-training objective, recent studies (Gao et al., 2021; Zhang et al.,

---
[1]https://github.com/luffy06/ReFusion

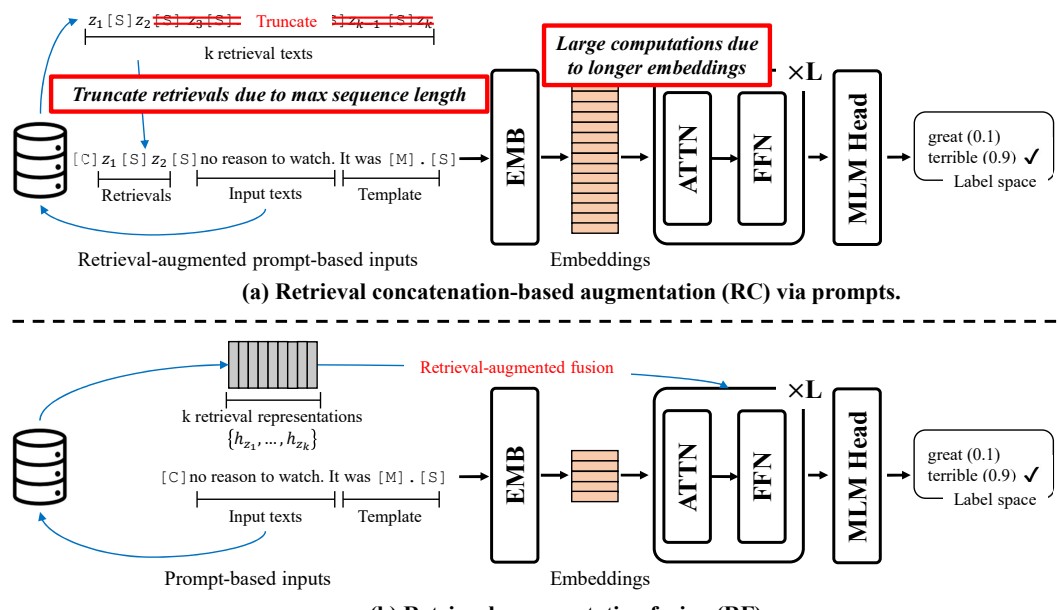

Figure 2: Comparisons between retrieval concatenation-based augmentation (RC) and retrieval representation fusion (RF).

2022; Guo et al., 2023b) suggest employing prompt-based fine-tuning. These works involve transforming the original inputs into prompt-based inputs incorporating a mask token for prediction. A simple example of prompt-based inputs of a single sentence or a pair of sentences is shown below,

$$X_{\texttt{prompt-single}} = \texttt{[CLS]}\, x \text{ It was } \texttt{[MASK]}\, . \, \texttt{[SEP]} \tag{1}$$

$$X_{\texttt{prompt-pair}} = \texttt{[CLS]}\, x_1 \, \texttt{[MASK]}\, ? \, x_2 \, \texttt{[SEP]} \tag{2}$$

The categorical label $y$ is transformed into semantic token $y_w$, e.g., using 'positive' to replace '1' and 'negative' to replace '0'. Then, the objective is to maximize the probability of the label word corresponding to the $\texttt{[MASK]}$ token,

$$Loss = -\log p(y_w | X_{\texttt{prompt}}) = -\log p(\texttt{[MASK]} = y_w | X_{\texttt{prompt}}) \tag{3}$$

To further improve prompt-based fine-tuning, recent works (Guo et al., 2023b; Chen et al., 2022) concatenate retrieval information as contexts or demonstrations of the given input, thus helping language models have a better semantic understanding. For example, as shown in Figure 2(a), they first retrieve top-$k$ similar sentences $Z = \{z_1, \ldots, z_k\}$ from an external key-value database for the given input $x$. Then, they concatenate all retrievals $Z$ with prompt-based inputs $X_{\texttt{prompt}}$. The retrieval-augmented prompt-based input of a single sentence is:

$$X_{\texttt{retrieval-single}} = \texttt{[CLS]}\, z_1 \, \texttt{[SEP]}\, \ldots \, \texttt{[SEP]}\, z_k \, \texttt{[SEP]}\, x \text{ It was } \texttt{[MASK]}\, . \, \texttt{[SEP]} \tag{4}$$

The objective is then optimized based on the retrieval-augmented prompt-based inputs.

**Motivations** Retrieval-augmented prompt-based fine-tuning trades off the amount of contextual information against the computational efficiency. Concatenating retrievals into the input enables language models to gather more contextual information, thus further improving language models' performance. However, due to the quadratic computational complexity of the attention mechanism, feeding a longer input would introduce a significant computational overhead. Besides, the limited length of the input is typically constrained by the hyper-parameter `max_length`, a longer input would be truncated before being fed in models (Figure 1(a)).

To quantify the impact of retrieval-based augmentations, this paper measures the performance of two baseline methods, i.e., Retrieval Concatenation-based augmentation (RC) and Retrieval-representation Fusion with addition (RF-add), in terms of the accuracy and the number of floating-point operations (FLOPs) in Figure 1(b). Specifically, RC concatenates retrievals with the given

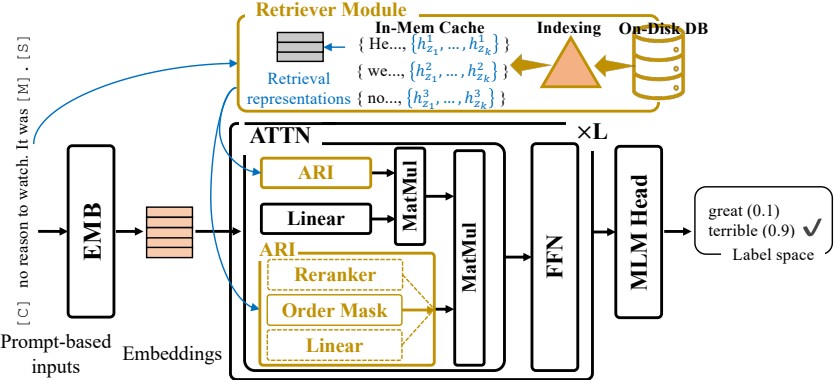

Figure 3: The architecture of the proposed ReFusion and the detailed structure of proposed modules.

input, while RF-add adds the retrieval representations to the hidden states of the `[CLS]` token [2]. Figure 1(b) demonstrates that concatenating more retrievals can slightly improve RC's accuracy while producing a significant amount of FLOPs. The degradation in RC's performance with 16 retrievals is due to the truncation of inputs, which results in a loss of semantic completeness. Conversely, when fusing more retrieval representations, the FLOPs of RF-add remains almost unchanged, yet its accuracy continues to exhibit an upward trend. Notably, RF-add achieves even higher accuracy than RC.

Consequently, this paper aims to propose a new paradigm of retrieval-based augmentation that directly integrates the retrieval representations into language models as shown in Figure 2(b). The intuition is to embed the knowledge directly into the thought process of models by merging models' hidden states with external information in an efficient fusion way. The preliminary experiments demonstrate the potential of the proposed paradigm. However, there are still some drawbacks to the baseline method (RF-add),

1. The external knowledge is obtained by a task-agnostic index based on a simple similarity metric, such as L2-Norm, rather than a more sophisticated semantic similarity metric;
2. RF-add pays the same attention to all retrievals on each representation dimension;
3. Not every layer of the model necessarily requires augmentation with retrievals.

These observations motivate us to propose a computation-efficient retrieval representation fusion with bi-level optimization.

## 3 ReFusion: A Computation-Efficient Retrieval Representation Fusion with Bilevel Optimization

In this section, we introduce the details of the proposed ReFusion framework. The ReFusion framework can be adapted to any transformer-based architecture (Vaswani et al., 2017), or any architecture that contains the attention module. As shown in Figure 3, we first present the retriever module used in the framework, which retrieves the representations of top-$k$ similar sentences. Then, we present the retrieval fusion module containing two ranking schemes, i.e., the reranker-based scheme and the ordered-mask scheme. Finally, we propose the adaptive retrieval integrator, which learns to choose the best combination of different ranking schemes across layers.

### 3.1 The Retriever Module

The retriever module includes a query encoder for encoding query texts, a task-agnostic index such as FAISS (Johnson et al., 2019) or ScaNN (Guo et al., 2020) built offline over billions of dense vectors, and a key-value store vector database for storing all text representations. The retrieving

---

[2]The term of the retrieval representations in this paper refers to the final layers' output vectors of retrievals. The term of the hidden states of a token refers to the intermediate vectors of the token.

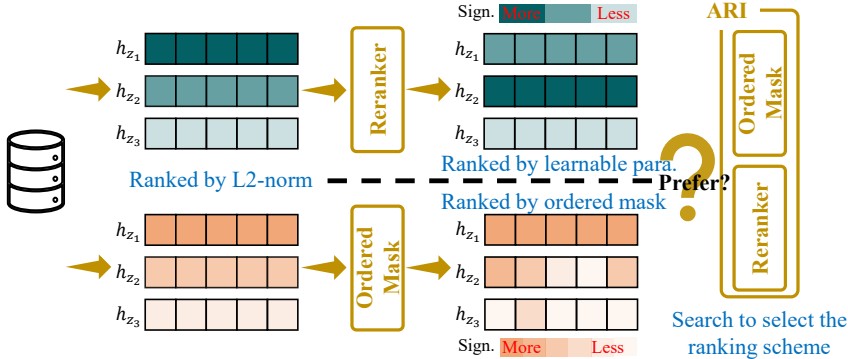

Figure 4: Two different ranking schemes used in the fusion module.

process in the framework is online performed, which means that for every forward, the query encoder first passes the representation $h_x$ of the input text $x$ to the index, then the index retrieves the representations $H_Z = \{h_{z_1}, \ldots, h_{z_k}\}$ of top-$k$ similar sentences $Z = \{z_1, \ldots, z_k\}$ to the fusion module.

## 3.2 THE RETRIEVAL FUSION MODULE

The retrieval fusion module first ranks the representations of top-$k$ similar sentences, then add them into the hidden states of existing modules. Specifically, this paper introduces two effective ranking schemes as shown in Figure 4.

### 3.2.1 RERANKING THE RETRIEVALS

In the retriever module, the retrievals are selected by a task-agnostic similarity metric, e.g., L2 norm. Directly adding the representations to the hidden states can only bring limited improvements. That is because 1) The ranking of retrievals should be different for each module layer in the model, as each module layer has different functionality (Jawahar et al., 2019); 2) The models should pay different attention to those retrievals in case of overemphasizing less relevant information. Therefore, this paper proposes a learnable reranker to learn the ranking distribution tailored to each module in the model. As shown in the top of Figure 4, the importance of retrievals is re-assigned after reranking.

Specifically, the reranker is $k$-dimensional learnable vector, i.e., $R = \{r_1, \ldots, r_k\}$, where $r_k \in \mathbb{R}$. The reranker is first normalized and then multiplied by the retrievals. Finally, the averaged representation of all reranked retrievals is added to the hidden states of the model, e.g., [CLS] token in BERT-like models (Liu et al., 2019b; Devlin et al., 2019). The formal steps are as follows,

$$r_i = \frac{\exp(r_i)}{\sum_j \exp(r_j)} \tag{5}$$

$$h^l_{y_{[\text{CLS}]}} = h^l_{x_{[\text{CLS}]}} + \frac{1}{k} \sum r_i \cdot h_{z_i} \tag{6}$$

where $h^l_{x_{[\text{CLS}]}}, h^l_{y_{[\text{CLS}]}}$ are the input and output hidden states of the [CLS] token in the $l$-th layer.

### 3.2.2 ORDERED MASK OVER RETRIEVAL REPRESENTATIONS

Rippel et al. (Rippel et al., 2014) proposed a nested dropout that directly drops the representation units from the sampled index $I$, thus yielding an inherent importance ranking on the representation dimensions. Based on the nested dropout, recent works (Cui et al., 2023; 2020; 2021; Mao et al., 2022) proposed the ordered mask that modeled the dropping process with a chain of Bernoulli variables and made it differentiable using the re-parameterization trick. Such methods rank the representation units for a better feature representation.

As shown in the bottom of Figure 4, this paper follows the main idea of the ordered mask but ranks the representation units over $k$ retrievals on each representation dimension. This is a more fine-grained ranking scheme compared to the previous ranking scheme. Specifically, let $h_{z_1}, \ldots, h_{z_k}$ be

the top-$k$ $D$-dimensional retrieval representations. For the $d$-th dimension of retrieval representation, the ordered mask is modeled by a chain of Bernoulli variables $V = \{v_1^d, \ldots, v_k^d\}$, where $v_i^d \sim$ **Bernoulli**$(\pi_i)$ indicates whether drop the $d$-th representation unit of the $i$-th retrieval. Following the property of nested dropout, the variable $v_i^d$ is conditioned on $v_{i-1}^d$; thus, we can obtain the marginal distribution $p(\mathbf{v_i^d})$.

After that, we use the re-parameterization trick, e.g., choosing the Gumbel Softmax distribution (Jang et al., 2017) as the tractable variational distribution $q(\mathbf{v_i^d})$. With Gumbel Softmax distribution, if $\mathbf{c}^d \sim$ **Gumbel**$(\beta, \tau)$, then $v_i^d = 1 - \text{cumsum}_i(\mathbf{c}^d)$, where $\mathbf{c}^d$ is a sample choice of the dropped index over $k$ retrievals on the dimension $d$, and $\text{cumsum}_i(\mathbf{c}^d) = \sum_{j=0}^{i-1} c_j^d$. In the Gumbel Softmax distribution, $\beta$ is a learnable parameter in the differentiable function $v_i^d = g(\epsilon_i; \beta)$ and $\tau$ is the temperature variable that controls the smoothness of the step at the dropped index.

Finally, we obtain $D$ different ordered masks $V^1, \ldots, V^D$ over the representation dimension. We use them to mask the retrievals in a fine-grained way. Then, the masked retrievals would be fused into the hidden states similarly. The formal steps are as follows,

$$\mathbf{c}^d \sim \textbf{Gumbel}(\beta, \tau) \tag{7}$$

$$v_i^d = 1 - \text{cumsum}_i(c^d) \tag{8}$$

$$\hat{h}_{z_i}^d = v_i^d \cdot h_{z_i}^d \tag{9}$$

$$h_{y_{\text{[CLS]}}}^l = h_{x_{\text{[CLS]}}}^l + \frac{1}{k} \sum \hat{h}_{z_i} \tag{10}$$

where $\hat{h}_{z_i}^d$ is the $d$-th masked representation unit of $i$-th retrieval.

### 3.3 THE ADAPTIVE RETRIEVAL INTEGRATOR

As shown in Figure 4, it is difficult to tell which ranking scheme is better on each module layer in the model. Therefore, this paper proposes the adaptive retrieval integrator (ARI) to find the optimal combination of ranking schemes across layers.

#### 3.3.1 SEARCH SPACE

This paper proposes to search for the combination of ranking schemes from two aspects, i.e., module level (whether it contains a retrieval fusion module in the module layer) and ranking scheme level (which ranking scheme is used for the module). For the module level, this paper replaces the linear module, e.g., query/key/value module in the attention, with the proposed adaptive retrieval integrator. For the ranking scheme level, the proposed adaptive retrieval integrator consists of two parallel retrieval fusion modules with different ranking schemes (e.g., the fusion module with the reranking scheme) and the original module without any ranking scheme.

Although the total number of candidates in the adaptive retrieval integrator is small (3 at most in this work), the whole search space is still quite large. Given a transformer-based language model with $N$ hidden layers, if only the key and value modules in every attention module are replaced, there are at least $(3 \times 3)^N = 9^N$ combinations of different ranking schemes. Taking the RoBERTa-large as an example (24 layers), the search space can be septillion-level large.

#### 3.3.2 SEARCHING WITH BI-LEVEL OPTIMIZATION

The searching challenge can be addressed by the bi-level optimization (Liu et al., 2019a). For the lower level, this paper aims to fine-tune the backbone model with the selected combination of ranking schemes on the training set. For the upper level, this paper aims to find the optimal combination of ranking schemes that maximizes performance on the validation set. Specifically, this paper creates $m$ architecture parameters in each adaptive retrieval integrator. For example, this paper uses $\alpha_{\text{key}}^l = \{\alpha_{\text{key},1}^l, \ldots, \alpha_{\text{key},m}^l\}$ to learn the choice for the key module in the $l$-th layer. To make the search space continuous, this paper also relaxes the categorical choice of a particular candidate to the softmax value over all possible candidates within each adaptive retrieval integrator,

$$\hat{o}_{\text{key}}^l(h) = \sum_i \frac{\exp(\alpha_{\text{key},i}^l)}{\sum_j \exp(\alpha_{\text{key},j}^l)} o_{\text{key},i}^l(h) \tag{11}$$

Table 1: Main results with RoBERTa-large.

| Methods | SST-2 | SST-5 | MR | CR | MPQA | SUBJ | TREC | CoLA | Avg-S |
|---|---|---|---|---|---|---|---|---|---|
| LM-BFF | $92.7_{0.9}$ | $47.4_{2.5}$ | $87.0_{1.2}$ | $90.3_{1.0}$ | $84.7_{2.2}$ | $91.2_{1.1}$ | $84.8_{5.1}$ | $9.3_{7.3}$ | 73.4 |
| DART | $\mathbf{93.5_{0.5}}$ | - | $\mathbf{88.2_{1.0}}$ | $\mathbf{91.8_{0.5}}$ | - | $90.7_{1.4}$ | $87.1_{3.8}$ | - | - |
| KPT | $90.3_{1.6}$ | - | $86.8_{1.8}$ | $88.8_{3.7}$ | - | - | - | - | - |
| CA-512 | $91.3_{1.4}$ | $46.7_{1.1}$ | $85.1_{1.4}$ | $88.3_{1.7}$ | $76.9_{2.8}$ | $88.0_{1.9}$ | $82.2_{4.4}$ | $7.4_{3.3}$ | 70.7 |
| ReFusion | $93.4_{0.6}$ | $\mathbf{49.8_{1.4}}$ | $87.9_{1.1}$ | $91.7_{0.3}$ | $\mathbf{86.7_{1.1}}$ | $\mathbf{92.5_{0.8}}$ | $\mathbf{90.3_{3.7}}$ | $\mathbf{11.4_{4.1}}$ | $\mathbf{75.5}$ |

| Methods | MNLI | MNLI-m | SNLI | QNLI | RTE | MRPC | QQP | Avg-P | Avg-all |
|---|---|---|---|---|---|---|---|---|---|
| LM-BFF | $68.3_{2.3}$ | $70.5_{1.9}$ | $77.2_{3.7}$ | $64.5_{4.2}$ | $69.1_{3.6}$ | $74.5_{5.3}$ | $65.5_{5.3}$ | 69.9 | 71.8 |
| DART | $67.5_{2.6}$ | - | $75.8_{1.6}$ | $66.7_{3.7}$ | - | $\mathbf{78.3_{4.5}}$ | $67.8_{3.2}$ | - | - |
| KPT | $61.4_{2.1}$ | - | - | $61.5_{2.8}$ | - | - | $\mathbf{71.6_{2.7}}$ | - | - |
| CA-512 | $66.2_{1.0}$ | $67.8_{1.3}$ | $71.6_{2.2}$ | $66.9_{3.2}$ | $66.6_{3.1}$ | $73.5_{6.9}$ | $64.0_{1.9}$ | 68.1 | 69.5 |
| ReFusion | $\mathbf{69.3_{1.5}}$ | $\mathbf{70.9_{1.5}}$ | $\mathbf{80.6_{1.4}}$ | $\mathbf{73.0_{1.1}}$ | $\mathbf{70.9_{2.3}}$ | $77.0_{3.6}$ | $68.9_{3.3}$ | $\mathbf{72.9}$ | $\mathbf{74.3}$ |

The results of LM-BFF, DART refer to their original paper. The results of KPT refer to (Chen et al., 2022).

where $o^l_{\text{key},i}(h)$ represents the output hidden states of the $i$-th candidate retrieval fusion module $o_i(\cdot)$, $\hat{o}(\cdot)$ indicates the overall output hidden states.

## 4 EXPERIMENT

In this section, we first introduce the dataset and the experimental setting. Then, we present the main results on 15 NKI tasks. We also give the ablation study to demonstrate the efficacy of each proposed module. Finally, we analyze the efficiency of the proposed ReFusion.

### 4.1 DATASETS AND EXPERIMENTAL SETTING

**Datasets** We conduct comprehensive experiments across 15 NKI tasks, including 8 tasks from GLUE benchmark(Wang et al., 2019), SNLI, SST-5, MR, CR, MNLI, MNLI-mm, Subj and TREC. There are 8 single-sentence tasks and 7 sentence-pair tasks. These tasks cover sentiment analysis, opinion polarity analysis, grammatical judgment, natural language inference, paraphrasing, etc. All these datasets' configurations are identical to that of LM-BFF (Gao et al., 2021).

**Experimental Settings** The proposed method was implemented using PyTorch framework, utilizing the computational power of two NVIDIA V100 GPUs. The experiments were conducted with the same settings as LM-BFF, which measures the average performance of five different sampled $D_{train}$ for each task with a fixed set of seed $S_{seed} = \{13, 21, 42, 87, 100\}$. In the dataset, there are 16 samples per class. The hyperparameters are listed as follows: the learning rate is 1e-5, the batch size is 32, the maximum sequence length is 128, the maximum steps are 1000, the number $k$ of similar sentences retrieved is set to 64 and we save the last checkpoint. We use AdamW as the optimizer. The models are based on RoBERTa-large for fair comparison with LM-BFF.

To validate the effectiveness, we compared ReFusion with several other models: (1) LM-BFF: a prompt-based fine-tuning approach; (2) DART(Zhang et al., 2022): a differentiable prompt-based model, that can automatically search for the optimal prompt; (3) KPT(Hu et al., 2022b): a prompt-based approach incorporating knowledge into the prompt verbalizer; and (4) CA-512: a retrieval-augmented prompt-based method concatenating retrievals with inputs. Unlike LM-BFF, DART, and KPT, we did not employ a grid search for the best parameters but instead chose a default template and parameters based on LM-BFF. Consequently, there is potential for the ReFusion to improve further through grid search. The templates are listed in Appendix A.

### 4.2 MAIN RESULTS

Table 1 presents the main experimental results of the ReFusion and comparisons on 15 NKI tasks. The results are shown in the form of means and variances, with the variance denoted by a subscript.

Table 2: Ablation studies on different modules.

| Methods | MPQA | SUBJ | TREC | SNLI | QNLI | RTE | Avg |
|---|---|---|---|---|---|---|---|
| Roberta-L | $83.6_{2.5}$ | $90.3_{2.8}$ | $83.8_{5.2}$ | $73.5_{5.2}$ | $65.0_{3.0}$ | $64.1_{2.0}$ | 76.7 |
| Reranker | $84.2_{2.2}$ | $91.3_{1.3}$ | $85.0_{4.2}$ | $74.3_{4.6}$ | $68.8_{1.4}$ | $65.6_{3.1}$ | 78.2 |
| Ordered Mask | $83.3_{1.9}$ | $90.8_{1.4}$ | $83.0_{5.8}$ | $74.9_{4.0}$ | $68.3_{1.4}$ | $65.8_{3.1}$ | 77.7 |
| ReFusion with Reranker | $86.9_{1.3}$ | $92.4_{1.3}$ | $\mathbf{90.8_{2.5}}$ | $80.3_{1.9}$ | $\mathbf{73.5_{1.8}}$ | $69.2_{2.4}$ | 82.2 |
| ReFusion with Ordered Mask | $\mathbf{87.0_{1.5}}$ | $92.4_{0.7}$ | $90.7_{3.0}$ | $80.3_{1.3}$ | $73.0_{1.0}$ | $70.4_{2.5}$ | **82.3** |
| ReFusion with All Rankings | $86.7_{1.1}$ | $\mathbf{92.5_{0.8}}$ | $90.3_{3.7}$ | $\mathbf{80.6_{1.4}}$ | $73.0_{1.1}$ | $\mathbf{70.9_{2.3}}$ | **82.3** |

For tasks with single sentences (S-Task), ReFusion consistently demonstrates superior performance across almost all benchmarks. ReFusion achieves state-of-the-art performance on 5 tasks over 8 tasks. Besides, ReFusion improves the average performance on the S-Task benchmark by about 2.1% than LM-BFF. Specifically, on the TREC task, ReFusion (90.3%) exhibits the maximum improvements over LM-BFF (84.8%). For tasks consisting of pair sentences (P-Task), ReFusion continues to demonstrate strong performance. ReFusion also achieves state-of-the-art results on 5 tasks over 7 tasks. ReFusion can improve the average performance on the P-Task benchmark by about 3.0% than LM-BFF. For instance, on the QNLI and SNLI benchmark, ReFusion (73% for QNLI, 80.6% for SNLI) significantly exceeds LM-BFF (64.5% for QNLI, 77.2% for SNLI).

The Avg-all represents the average performance of all 15 NKI tasks. For overall average performance, ReFusion achieves a score of 74.3%, marginally surpassing LM-BFF's 71.8%. This further highlights ReFusion's consistent and superior performance. Besides, ReFusion surpasses other models like DART, CA-512, and KPT, delivering superior or comparable results. Notably, the standard deviation of ReFusion is considerably smaller than that of other models, indicating that ReFusion produces stable results and offers superior robustness.

## 4.3 ABLATION STUDY

We conduct ablation experiments on six representative tasks to show the contributions of each module to the overall performance. As shown in Table 2, compared to the baseline method (Roberta-large), adding ranked retrieval representations into hidden states (lines 2-3) can slightly improve the model's performance. However, the improvement is limited as the ranking schemes may not always be suitable for every module in the model. With the adaptive retrieval integrator searching for the best combination of different ranking schemes, those methods using the ReFusion framework (lines 4-6) achieve significant improvements compared to those baselines (lines 1-3). The ReFusion searching among all ranking schemes (ReFusion with All Rankings) achieves the best performance. This indicates that the adaptive retrieval integrator can learn the functionality of each module and choose the most suitable ranking schemes for each module.

## 4.4 ANALYSIS

In this section, we analyze the ReFusion in terms of efficiency. As shown in the Table 3, we report the model performance and the inference latency based on the different queries, e.g., 'Input' which retrieves based on the input texts and 'Hidden' which retrieves based on the hidden states. Experimental results show that retrieving based on the hidden states can further improve the model performance but at the expense of increasing computational times. The benefit of 'Hidden' is that the retrieval representations can dynamically change as hidden states change. However, 'Hidden' requires retrieving at each retrieval fusion module, which may be only suitable for scenarios where adaptability is more important than time constraints, particularly in offline contexts.

Table 4 shows the latency breakdown of the model forward process, including the baseline model (Roberta-large) and the retrieval-augmented model (CA-512 and ReFusion). The inference latency of the baseline model is about 7.38 seconds per sentence. For retrieval concatenation-based augmentation (CA-512), the time cost of the retrieving process is about 6.63 seconds. However, due to the long input length and the quadratic complexity of attention, the time cost of the model forward significantly increases (31.93 seconds), thus leading to a longer inference latency. In the proposed

Table 3: The trade-off between model performance and efficiency. 'Input' and 'Hidden' refer to retrieving knowledge based on input texts and hidden states, respectively. 'Acc' and 'Latcy' refer to the accuracy of the task and the inference latency per sentence in seconds, respectively.

| Queries | | MPQA | SUBJ | TREC |
|---|---|---|---|---|
| Input | Acc | 87.9 | **92.1** | 84.0 |
| | Latcy | 3.7 | 3.8 | 11.1 |
| Hidden | Acc | **88.7** | 91.7 | **85.6** |
| | Latcy | 108.2 | 109.4 | 294.9 |

Table 4: Latency breakdown of model forward. 'F', 'R', and 'All' refer to the latency of the model forward, retrieving, and overall latency, respectively. All results are shown in seconds. More results can be found in the Appendix.

| Methods | | MPQA | SUBJ | TREC |
|---|---|---|---|---|
| Roberta-L | F | 7.35 | 7.41 | 7.37 |
| | R | 5.71 | 8.46 | 5.72 |
| CA-512 | F | 31.69 | 32.24 | 31.86 |
| | All | 37.40 | 40.70 | 37.58 |
| | R | 11.27 | 13.56 | 11.43 |
| ReFusion | F | 8.95 | 9.03 | 9.02 |
| | All | 20.22 | 22.59 | 20.45 |

ReFusion, the number of retrievals is not limited by the hyperparameter of `max_length`, the retriever module can retrieve more related knowledge at the cost of a bit more time (12.09 seconds). As expected, the inference latency of the model forward in the ReFusion is only a bit longer than the baseline due to the time cost of retrieval fusion modules. Overall, the proposed ReFusion makes a better trade-off between the model performance and the model efficiency for retrieval-based augmentation.

## 5 RELATED WORK

**Retrieval-Augmented Transformers** Retrieval-augmented transformers (Borgeaud et al., 2022b; Lewis et al., 2020; Shi et al., 2022; Zhou et al., 2023; Frisoni et al., 2022) leverage the information retrieval techniques to augment modern transformer-based language models with external knowledge databases. These retrieval-based augmentations can be classified into two types of work, i.e., retrieval concatenation-based augmentation (RC) and retrieval representation fusion (RF). The RC generally concatenates retrieval texts or representations. Ori Ram et al. (Ram et al., 2023) concatenated retrievals as the context of inputs, thus providing models with more information. Different from this, other works (Izacard & Grave, 2021; Guo et al., 2023a) first leveraged encoders to encode retrievals, then fed the feature representation concatenated by the retrieval representations and the input representation into a decoder. Those techniques will result in a long sequence and then introduce large computations in the attention module. The RF involves introducing retrieval representations inside models. Some works (Borgeaud et al., 2022a; Li et al., 2022) proposed to fuse the retrievals using a newly added cross-attention module. However, those works require pre-training the newly added cross-attention module and the backbone model. Besides, the cross-attention module results in large computational overheads. The ReFusion in this paper introduces a computation-efficient retrieval fusion module that can be fine-tuned with a few-shot dataset.

## 6 CONCLUSION

This paper proposes a new paradigm of retrieval-based augmentations, solving the bottleneck of retrieval concatenation-based augmentations. This paper proposes a computation-efficient retrieval representation fusion framework with bi-level optimization named ReFusion. Experimental results demonstrate that ReFusion achieves superior performance while saving considerable computational resources. For the future work, we plan to investigate more ranking schemes. Besides, it is also worthwhile to explore the effect of different hyper-parameters in the framework, such as the module to fuse retrievals and the number of retrieved representations.

## 7 ACKNOWLEDGEMENT

The work described in this paper was supported by a grant from the Research Grants Council of the Hong Kong Special Administrative Region, China (Project No. CityU 11209122).

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

## A    TEMPLATES ON ALL TASKS

Table 5 provides an overview of the manual templates and selected label words used for each dataset in this paper. These templates and label words were created following LM-BFF (Gao et al., 2021).

Table 5: Templates and label words used in this paper.

| Task | Prompts | Label word |
|------|---------|------------|
| SST-2 | `[CLS]` $x$ It was `[MASK]`. `[SEP]` | "0":"terrible", "1":"great" |
| SST-5 | `[CLS]` $x$ It was `[MASK]`. `[SEP]` | "0":"terrible","1": "bad", "2": "okay","3": "good","4": "great" |
| MR | `[CLS]` $x$ It was `[MASK]`. `[SEP]` | "0":"terrible", "1":"great" |
| CR | `[CLS]` $x$ It was `[MASK]`. `[SEP]` | "0":"terrible", "1":"great" |
| MPQA | `[CLS]` $x$ It was `[MASK]`. `[SEP]` | "0":"terrible", "1":"great" |
| SUBJ | `[CLS]` $x$ This is `[MASK]`. `[SEP]` | "0":"subjective", "1":"objective" |
| TREC | `[CLS]` `[MASK]` $x$ `[SEP]` | "0":"Description","1":"Entity","2":"Expression", "3":"Human","4":"Location","5":"Number" |
| CoLA | `[CLS]` $x$ It was `[MASK]`. `[SEP]` | "0":"incorrect", "1":"correct" |
| MNLI | `[CLS]` $x_1$ ? `[MASK]`, $x_2$ `[SEP]` | "contradiction": "No","entailment":"Yes", "neutral": "Maybe" |
| MNLI-m | `[CLS]` $x_1$ ? `[MASK]`, $x_2$ `[SEP]` | "contradiction": "No","entailment":"Yes", "neutral": "Maybe" |
| SNLI | `[CLS]` $x_1$ ? `[MASK]`, $x_2$ `[SEP]` | "contradiction": "No","entailment":"Yes", "neutral": "Maybe" |
| QNLI | `[CLS]` $x_1$ ? `[MASK]`, $x_2$ `[SEP]` | "not entailment":"No ","entailment":"Yes" |
| RTE | `[CLS]` $x_1$ ? `[MASK]`, $x_2$ `[SEP]` | "not entailment":"No ","entailment":"Yes" |
| MRPC | `[CLS]` $x_1$ `[MASK]`, $x_2$ `[SEP]` | "0":"No", "1":"Yes" |
| QQP | `[CLS]` $x_1$ `[MASK]`, $x_2$ `[SEP]` | "0":"No", "1":"Yes" |

## B    RESULTS ON AUTOREGRESSIVE MODELS

This paper also evaluates the ReFusion with the classical autoregressive language model T5 Raffel et al. (2020). As shown in Table 6, ReFusion outperforms the T5-base model on most tasks, raising the overall average score from 78.2 to 79.6. The results are consistent with that of Roberta-large, demonstrating the robustness of ReFusion and the effectiveness of applying ReFusion on transformer-based models.

Table 6: Results with T5-base.

| Methods | SST-2 | SST-5 | MR | CR | MPQA | SUBJ | TREC | CoLA | Avg-S |
|---------|-------|-------|----|----|----|------|------|------|-------|
| T5-base | $91.0_{0.3}$ | $46.6_{1.7}$ | $87.6_{1.0}$ | $\mathbf{91.4_{0.2}}$ | $85.9_{1.2}$ | $84.5_{3.7}$ | $91.8_{5.0}$ | $51.1_{3.3}$ | 78.8 |
| ReFusion (T5) | $\mathbf{92.4_{0.3}}$ | $\mathbf{47.2_{0.9}}$ | $\mathbf{88.3_{0.9}}$ | $\mathbf{91.4_{0.4}}$ | $\mathbf{86.1_{1.5}}$ | $\mathbf{87.4_{1.3}}$ | $\mathbf{94.9_{0.5}}$ | $\mathbf{53.7_{4.1}}$ | **80.2** |

| Methods | MNLI | MNLI-m | SNLI | QNLI | RTE | MRPC | QQP | Avg-P | Avg-all |
|---------|------|--------|------|------|-----|------|-----|-------|---------|
| T5-base | $73.9_{0.8}$ | $75.8_{0.6}$ | $79.9_{2.1}$ | $82.1_{1.9}$ | $79.0_{1.9}$ | $75.5_{3.6}$ | $\mathbf{76.2_{0.5}}$ | 77.5 | 78.2 |
| ReFusion (T5) | $\mathbf{75.3_{1.1}}$ | $\mathbf{77.3_{1.0}}$ | $\mathbf{83.0_{0.8}}$ | $\mathbf{83.6_{0.5}}$ | $\mathbf{79.1_{1.6}}$ | $\mathbf{77.8_{2.7}}$ | $76.1_{0.8}$ | **78.9** | **79.6** |

## C    RESULTS ON FULL TRAINING SET

This paper compares the proposed ReFusion with LM-BFF (Gao et al., 2021) on several tasks under the prompt-based setting with the full training set. As shown in Table 7, ReFusion generally demonstrates superior performance compared to LM-BFF. The average performance of ReFusion surpasses that of LM-BFF by 0.9%. The results suggest that ReFusion's performance superiority is consistent and not dependent on the size of the dataset. This also implies that ReFusion is robust and can generalize well across varying amounts of data.

Table 7: Full training set results compared with LM-BFF.

| Methods | SST-2 | SST-5 | MR | CR | MPQA | SUBJ | TREC | CoLA | RTE | Avg |
|---------|-------|-------|-----|-----|------|------|------|------|------|------|
| LM-BFF | 95.0 | 58.7 | 90.8 | 89.4 | **87.8** | 97.0 | 97.4 | 62.6 | 80.9 | 84.6 |
| ReFusion | **95.6** | **61.0** | **92.3** | **91.4** | 84.4 | **97.1** | **97.6** | **62.8** | **85.2** | **85.3** |

## D  THE MODULE CHOICE WHERE THE RETRIEVALS ARE FUSED

Table 8 presents the results of fusing the retrievals into different modules, i.e., key, value, query, and ffn (feed-forward network) modules. Experimental results show that fusing retrievals into only key or query, key, and value modules can achieve the best performance. However, except for the ffn module, fusing retrievals into the modules in the attention modules yields competitive performance. This indicates that the proposed ReFusion can improve the model's focus.

Table 8: Different module choices where the retrievals are fused.

| Module | MPQA | SUBJ | TREC | SNLI | QNLI | RTE | Avg |
|--------|------|------|------|------|------|------|------|
| key | 86.7 | 93.5 | **94.2** | 79.2 | **76.6** | 71.1 | **83.6** |
| value | 86.6 | 93.4 | 94.0 | 78.5 | 76.3 | 71.8 | 83.4 |
| query | **87.3** | 93.6 | 93.6 | 78.6 | 76.3 | 71.8 | 83.5 |
| key and value | 86.3 | 93.3 | 93.0 | **81.0** | 72.1 | **73.3** | 83.2 |
| query and key | 86.6 | 93.6 | **94.2** | 78.6 | 74.8 | 72.6 | 83.4 |
| query and value | 86.7 | **93.9** | 93.0 | 80.1 | 74.8 | 72.6 | 83.5 |
| query and key and value | 86.3 | 92.9 | **94.2** | 79.2 | 76.4 | 72.6 | **83.6** |
| ffn | 84.0 | 93.7 | **94.2** | 78.0 | 75.5 | 70.4 | 82.6 |

## E  IMPACT OF RETRIEVER METRICS

The retriever metrics measure the similarity of different data in the retrieval database. Table 9 shows the results of the ReFusion retrieving based on two similarity metrics. Experimental results show that using the inner product has a slightly higher overall performance than that of the L2 norm. This observation suggests the potential for further enhancements in ReFusion through the adoption of inner product metrics.

Table 9: The impact of retriever metrics. 'L2' and 'IP' refer to the metric of L2-norm and inner-product.

| Metrics | MPQA | SUBJ | TREC | SNLI | QNLI | RTE | Avg |
|---------|------|------|------|------|------|------|------|
| L2 | **86.7** | **92.5** | 90.3 | 80.6 | 73.0 | 70.9 | 82.3 |
| IP | 86.4 | 92.1 | **91.2** | **80.9** | **73.3** | **71.8** | **82.6** |

## F  IMPACT OF RETRIEVAL NUMBER $k$

Table 10 illustrates the impact of varying the number of retrievals. As $k$ increases from 1 to 8, the proposed ReFusion improves performance on most tasks. This suggests that fusing more retrievals can add valuable information to enhance the model's performance. As $k$ continues to increase, there is a slight fluctuation in performance. For example, the TREC shows a peak performance at $k = 16$, and the SNLI peaks at $k = 32$. The average performance remains relatively stable, around 83.5 to 83.8. When $k$ is larger than 64, there is a general trend of either plateauing or slightly declining performance across most tasks. This could imply that there may be diminishing returns with too many retrievals, with additional information possibly introducing noise or irrelevant data that complicating the model's filtering process.

Table 10: The impact of retrieval number $k$.

| $k$ | MPQA | SUBJ | TREC | SNLI | QNLI | RTE | Avg |
|---|---|---|---|---|---|---|---|
| 1 | 86.7 | 92.6 | 91.0 | 78.2 | 74.3 | **72.9** | 82.6 |
| 2 | 85.9 | 93.6 | 90.2 | 79.4 | 74.8 | 71.5 | 82.6 |
| 4 | 86.1 | 93.4 | 93.8 | 79.6 | 74.6 | 72.2 | 83.3 |
| 8 | **87.8** | **93.8** | 93.0 | 79.1 | 74.9 | 72.6 | 83.5 |
| 16 | 87.0 | **93.8** | **94.4** | 79.8 | **75.3** | 71.5 | 83.6 |
| 32 | **87.8** | 92.9 | **94.4** | **81.1** | 74.6 | 72.2 | **83.8** |
| 64 | 86.3 | 93.3 | 93.0 | 79.6 | 74.7 | 71.8 | 83.1 |
| 128 | 86.6 | 93.0 | 93.6 | 79.4 | 74.9 | 72.2 | 83.3 |
| 256 | 85.1 | 92.4 | 93.4 | 80.4 | 74.5 | 72.2 | 83.0 |
| 512 | 87.0 | **93.8** | 93.2 | 80.2 | 74.9 | 72.2 | 83.6 |

## G  RESULTS ON FEW-SHOT KNOWLEDGE-INTENSIVE TASKS.

This paper also evaluates the ReFusion on knowledge-intensive tasks, such as the SQuAd dataset, in the few-shot setting (16-shot). Table 11 shows that the ReFusion significantly enhances the baseline model's performance, outperforming other comparison methods. This suggests the ReFusion substantially contributes to the model's understanding and reasoning abilities in knowledge-intensive tasks.

Table 11: Results on few-shot knowledge-intensive tasks.

| Methods | SQuAD |
|---|---|
| T5-base | $46.6_{3.9}$ |
| Splinter (Ram et al., 2021) | 54.6 |
| FewshotBART (Chada & Natarajan, 2021) | $55.5_{2.0}$ |
| ReFusion (T5) | $\mathbf{58.6_{0.9}}$ |

## H  RESULTS WITH LoRA.

This paper also evaluates the ReFusion with different fine-tuning techniques, such as LoRA (Hu et al., 2022a). Table 12 shows the results of using LoRA to fine-tune the backbone model with/without the ReFusion. Generally, the ReFusion can also outperform the baseline from 63.8 to 64.7. However, compared to full parameters fine-tuning, the degree of improvements in Table 12 is not competitive. This may lie in the fact that in the few-shot setting, fine-tuning the model with LoRA and RA on NKI tasks is quite a difficult task for deep learning models, requiring stronger learning and alignment capability, which is still a promising domain to be explored.

Table 12: Results with Roberat-large tuned by LoRA (Hu et al., 2022a).

| Methods | SST-2 | SST-5 | MR | CR | MPQA | SUBJ | TREC | CoLA | Avg-S |
|---|---|---|---|---|---|---|---|---|---|
| Roberta-L | $\mathbf{91.4_{0.1}}$ | $\mathbf{45.2_{4.3}}$ | $87.5_{0.3}$ | $90.2_{0.9}$ | $78.9_{2.3}$ | $85.3_{0.7}$ | $41.4_{0.3}$ | $\mathbf{4.0_{1.8}}$ | 65.5 |
| ReFusion | $91.1_{0.7}$ | $44.8_{0.8}$ | $\mathbf{88.5_{0.4}}$ | $\mathbf{90.8_{0.1}}$ | $\mathbf{81.1_{0.6}}$ | $\mathbf{89.1_{0.6}}$ | $\mathbf{46.5_{4.9}}$ | $2.5_{0.2}$ | **66.8** |

| Methods | MNLI | MNLI-m | SNLI | QNLI | RTE | MRPC | QQP | Avg-P | Avg-all |
|---|---|---|---|---|---|---|---|---|---|
| Roberta-L | $\mathbf{57.0_{0.1}}$ | $\mathbf{57.4_{0.6}}$ | $\mathbf{68.9_{5.2}}$ | $62.4_{0.5}$ | $\mathbf{56.7_{3.0}}$ | $\mathbf{72.3_{6.4}}$ | $59.4_{1.3}$ | 62.0 | 63.8 |
| ReFusion | $56.8_{0.1}$ | $57.3_{0.8}$ | $67.8_{0.6}$ | $\mathbf{64.2_{1.1}}$ | $56.5_{1.8}$ | $71.2_{3.7}$ | $\mathbf{62.4_{3.7}}$ | **62.3** | **64.7** |

