# OpenReview forum: "ReFusion: Improving Natural Language Understanding with Computation-Efficient Retrieval Representation Fusion"
_ICLR.cc/2024/Conference — ICLR 2024 poster_

### Official Review · Reviewer_UAxr · 2023-10-31

**Soundness:** 4 excellent
**Presentation:** 3 good
**Contribution:** 3 good
**Rating:** 8
**Confidence:** 3

**Summary:**

This paper develops a new method, ReFusion, for combining retrieved contextual representations with example representations to improve models of non-knowledge intensive tasks like sentiment and NLI. The approach adds a weighted average of retrieved passage representations into the CLS token representations. These representations can be ranked by a learned weighting scheme or a learned mask over dimensions. For ReFusion, this choice is made by a search module. Overall, ReFusion does well in evaluations, and the paper includes valuable ablation studies that help us understand which pieces contribute most. Those studies suggest that the search component is especially important.

**Strengths:**

This is a thoughtful paper that poses a well-defined, tightly circumscribed problem and addresses it in a creative and successful way. Assuming we are working with models with small window sizes, this seems to be a good way to handle lots of context. It's also valuable to have additional evidence that retrieval (in the form of additional similar examples) helps even for tasks like the ones addressed here.

**Weaknesses:**

1. The argument for keeping prompts short is growing less compelling as models can handle more and more context. The other savings from the current approach seem to be small, since we still need to create representations for every contextual element.

2. In terms of clarity: overall, the paper is good, and the methods are mostly easy to follow, but I am not sure how the architecture search works. Section 3.3.1 creates on set of expectations, and section 3.3.2 does not seem to meet those expectations, or even respond to them directly. In particular, we are led to believe that the search module will decide what method to use on a per-layer basis, but then it seems to turn out to be a global choice. But I many well simply not be understanding the description and its connection to the experimental results.

**Questions:**

Could you provide the details on how many train examples the various methods represented in Table 1 need, either as supervised examples or as demonstrations? I think this is a significant favor in how models perform on the chosen tasks. If all of them use all the train examples, that suffices as an indication that the playing field is level. (I am not asking for an efficiency analysis or anything here.)

---

> ### Comment · Reviewer_UAxr · 2023-11-21
> **Eager for responses to my question**
>
> I gave a high score, but I am still eager to have my question about training data addressed. This really matters to me when it comes to assessing the experimental evidence.

---

> ### Author Response · Authors · 2023-11-21
>
> **W1**: Thanks for your suggestions. Although existing modern language models can handle quite a long input sequence (e.g., 128k for GPT-4-turbo), the resource overhead for self-attention is still non-negligible, i.e., larger memory requirements as well as computational costs. When the long input sequence is due to retrievals, our ReFusion can significantly reduce the resource overhead, maintaining almost the same FLOPs (from 188.6541 GFLOPs to 188.6548 GFLOPs shown in Figure 1). For other savings, the retrieval representations are encoded by a frozen pre-trained language model and stored in a task-agnostic key-value database. These processes are done offline before the training/inference. During the fusion process, our ReFusion only retrieves the pre-built representations and directly fuses them into the language models, thus introducing few overheads.
>
> **W2**: Thanks for your suggestions. We recognize that there are some unclear descriptions in Section 3.3.2, especially the presentation of notations. We will carefully revise the Section 3.3. The hierarchy of architecture search can be classified into three levels, i.e., network level (global), layer level, module level. We are working on the finest granularity optimization (module level). Specifically, we replace several original modules with our search modules per layer.
>
> To make it clear, we reformulate the notations used in Section 3.3.2.
> * The superscript $l$ indicates the index of the layer.
> * The subscript $\text{key/value/query}$ indicates the module our search module will replace.
> * The subscript $m$ indicates the number of candidates used in each search module.
>
> Taking the key module of the $l$-th layer as an example, our search module contains three ranking scheme candidates (i.e., $m=3$) and a vector of architecture parameters, i.e., $\alpha\_{\text{key}}^l=\left( \alpha\_{\text{key},1}^l,\alpha\_{\text{key},2}^l,\alpha\_{\text{key},3}^l \right)$. Then, we use a softmax operation over all candidates within the search module to relax the categorical choice,
>
> $$
> \hat{o}\_{\text{key}}^l (h)=\sum\_i \frac{\exp⁡{\alpha\_{\text{key},i}^l}}{\sum\_j \exp⁡{\alpha\_{\text{key},j}^l}} o\_{\text{key},i}^l (h)
> $$
>
> where $o\_{\text{key},i}^l (h)$ represents the output of the $i$-th candidate within the search module which replaces the key module in the $l$-th layer, $\hat{o}\_{\text{key}}^l (h)$ represents the final output of the search module that replaces the key module in the $l$-th layer. Finally, we use bilevel optimization to jointly optimize the architecture parameters and model weights.
>
> **Q1**: Thanks for your comments. The number of training examples used in Table 1 for all baselines as well as our ReFusion is 16 per class, which is the experimental setting used in LM-BFF. For example, the task SST-2 has two classes, thus we use 32 training examples. For fair comparisons, we don’t add any examples as demonstrations for either LM-BFF or our ReFusion.

---

### Official Review · Reviewer_UsSB · 2023-11-01

**Soundness:** 3 good
**Presentation:** 3 good
**Contribution:** 2 fair
**Rating:** 6
**Confidence:** 3

**Summary:**

Retrieval-augmented models have gained much attention for knowledge-intensive tasks, yet the core drawback is the increased computation brought by the augmented context. The paper proposes retrieval-augmented natural language understanding model that is computation-efficient. Specifically, the authors introduces three modules: 1) retrieval,  2) reranker, and 3) architecture search module. The retrieval module retrieves relevant sentences, then reranker reweighs retrieved representation, and finally, search module finds suitable fusion module for each layer. At inference, the search module simply picks a module with the highest score and utilizes the module for inference.

**Strengths:**

- The authors illustrate the effectiveness of the proposed model on 15 NLU tasks.
- The proposed model is technically sounding. Injecting retrievals via representation fusion is reasonable.
- Experimental settings are well listed out for reproducibility
- Appendix covers meaningful discussions, such as the effect of retrieval with different k

**Weaknesses:**

- Some terms are confusing. The authors use the term "neural architecture search" for the search module, but rather, the search module simply performs hard gate operation among the fusion modules.
- Notations can be improved. When the authors write "h_y_{<cls>}", this seems like the output of the final layer. However, the authors perform fusion at every layer. Therefore, I suggest the author to denote layer with superscript l.
- The authors stress "computation-efficient" part, but it would be interesting to see the actual throughput and speed compared to other baselines methods as the proposed method brings in multiple modules.
- The authors should add information on the number of parameters. It is not clear whether the empirical gains are obtained by simply adding more learnable parameters compared to that of LM-BFF.

**Questions:**

Q1. I don't get how section 3.2.1 works. The authors states that "reranker is a 1D learnable vectors of k dimensions". If so, each rank (decided by retriever) is assigned with a learned scalar. This means that the learned 1D vector is shared among all the retrievals. Could you clarify on this point?

---

> ### Author Response · Authors · 2023-11-22
>
> **W1**: Thanks for your suggestions. We acknowledge the concern regarding the use of the term “neural architecture search” and recognize that using this term may imply a broader scope of architecture search than what is used in our search module. We will carefully revise the manuscript and use the term “Adaptive Retrieval Integrator (ARI)” to specify our methods in the search module.
>
> For the architecture search module, we first define the search space and then describe the details about how to find the optimal architecture using bilevel optimization. The search space is basically determined by two factors, i.e., the number of ranking scheme candidates and the number of original modules that will be replaced by our search module. Within our search module, there are multiple ranking scheme candidates with a learnable architecture weight. We use bilevel optimization to train the search process. On the upper level, we aim to find the optimal combination of ranking schemes that maximizes performance on a validation set. During this level, we only update the architecture weights. On the lower level, we aim to fine-tune the model given the current selection of ranking schemes from the upper level. This involves minimizing the loss of downstream tasks. For the lower level, we only update model’s weights on a training set.
>
> **W2**: Thanks for your suggestions. We revised the notations with the superscript that denotes the layer index.
>
> **W3**: Thanks for your comments. We evaluate the inference latency per sentence of our ReFusion, the baseline LM-BFF, and the retrieval-augmented baseline CA-512 on 15 NKI tasks. Our ReFusion can significantly accelerate the inference efficiency compared to CA-512, and makes a better trade-off between performance and efficiency compared to LM-BFF. Please refer to the Table 1 in the response to the **Reviewer 9abu** for more details.
>
> **W4**: Thanks for your suggestions. We counted the number of parameters of LM-BFF and our ReFusion during the training stage in Table 1. Compared to LM-BFF, our ReFusion only increases the number of parameters by 0.88%, which is quite trivial. However, our ReFusion can improve the performance by about 2.5 on average.
>
> Table 1. The number of parameters of ReFusion and LM-BFF.
> | **Methods** | **# Params (M)** |
> | --- | --- |
> | LM-BFF | 357.51 |
> | ReFusion | 360.66 |
>
> **Q1**: Thanks for your comments. The reranker is a $k$-dim weight vector, where the $i$-th value in vector indicates the task-related importance of the $i$-th retrieval. The reranker is designed to learn the mappings between task-related importance and task-agnostic similarity, rather than the specific importance of retrievals. For example, ReFusion uses L2 distance to retrieve the top-$k$ nearest neighbors. But these retrievals might not be semantically suitable for downstream tasks. Thus, ReFusion adopts a simple $k$-dim vector to re-weight the importance of retrievals for the downstream tasks.

---

### Official Review · Reviewer_aM36 · 2023-11-09

**Soundness:** 3 good
**Presentation:** 3 good
**Contribution:** 3 good
**Rating:** 3
**Confidence:** 4

**Summary:**

The paper discusses an approach to enhance prompt-based fine-tuning techniques for language models by integrating retrieval-augmented methods. The main contribution is the introduction of the ReFusion framework, which fuses representations of retrieved similar texts directly into the model to address performance and efficiency bottlenecks in existing techniques.

**Strengths:**

1. ReFusion introduces a novel approach to combining retrieval information with model representations, which is a step forward in prompt-based learning.
2. The experimental results indicate that ReFusion achieves superior performance over other models, suggesting a better understanding capability of the language models.
3. Evaluation comprehensivenss: The paper conducts a series of experiments on 15 different natural language understanding (NKI) tasks, which is a broad evaluation of the ReFusion method.

**Weaknesses:**

1. Scope of Experiments: The research confines its experimentation to masked language models, leaving the effectiveness of the proposed ReFusion method on populr autoregressive language models unexplored. Clarification on its adaptability to such models would be beneficial for broader application.

2. Performance on Knowledge-Intensive Tasks: While the paper presents improved results for non-knowledge-intensive tasks using ReFusion, it remains unclear how this method stacks up against others in knowledge-intensive scenarios. Given that retrieval-augmented language models are often specifically leveraged for their prowess in knowledge-intensive tasks, a direct comparison in this primary use case would be valuable.

3. Compatibility with Current Language Models: Modern language models, such as GPT-4 and LongChat, are designed to handle extended sequences, which facilitates the use of context augmentation methods. However, the representation fusion approach proposed by ReFusion necessitates fine-tuning of language models. This requirement may not be compatible with the current trend of employing language models as 'black-box' functions, where fine-tuning is either not possible or practical due to access or resource constraints.

**Questions:**

1. The caption for Figure 2 is not clear.
2. The discussion on related works in the paper does not appear to be exhaustive. For example, Important references, such as the context-augmentation method and retrieval representation fusion, are absent.

---

> ### Author Response · Authors · 2023-11-17
>
> **W1**: Thanks for your suggestions. We applied our ReFusion to the popular autoregressive language models, such as T5 [1]. As shown in Table 1, we evaluate ReFusion (T5-base) on 15 NKI tasks with 5 random seeds. Experimental results show that our ReFusion can be adapted to autoregressive models, improving the model performance by 1.4 on average.
>
> Table 1. Results with T5-Base.
> | **Methods**        | **SST-2** | **SST-5** | **MR**   | **CR**   | **MPQA** | **Subj** | **TREC** | **CoLA** | **Avg-S** |
> |--------------------|-----------|-----------|----------|----------|----------|----------|----------|----------|-----------|
> | T5-Base            | ${91.9}_{0.3}$      | ${46.6}_{1.7}$      | ${87.6}_{1.0}$     | ${91.4}_{0.2}$     | ${85.9}_{1.2}$     | ${84.5}_{3.7}$     | ${91.8}_{5.0}$     | ${51.1}_{3.3}$     | ${78.8}_{0.7}$      |
> | ReFusion (T5-Base) | $\boldsymbol{92.4}_\boldsymbol{0.3}$  | $\boldsymbol{47.2}_\boldsymbol{0.9}$  | $\boldsymbol{88.3}_{0.9}$ | $\boldsymbol{91.4}_\boldsymbol{0.4}$ | $\boldsymbol{86.1}_\boldsymbol{1.5}$ | $\boldsymbol{87.4}_\boldsymbol{1.3}$ | $\boldsymbol{94.9}_\boldsymbol{0.5}$ | $\boldsymbol{53.7}_\boldsymbol{4.1}$ | $\boldsymbol{80.2}_\boldsymbol{0.6}$  |
>
> Table 1 (cond.). Results with T5-Base.
> | **Methods**        | **MNLI** | **MNLI-mm** | **SNLI** | **QNLI** | **RTE**  | **MRPC** | **QQP**  | **Avg-P** | **Avg-all** |
> |--------------------|----------|-------------|----------|----------|----------|----------|----------|-----------|-------------|
> | T5-Base            | ${73.9}_{0.8}$     | ${75.8}_{0.6}$        | ${79.9}_{2.1}$     | ${82.1}_{1.9}$     | ${79.0}_{1.9}$     | ${75.5}_{3.6}$     | $\boldsymbol{76.2}_\boldsymbol{0.5}$ | ${77.5}_{0.3}$      | ${78.2}_{0.4}$        |
> | ReFusion (T5-Base) | $\boldsymbol{75.3}_\boldsymbol{1.1}$ | $\boldsymbol{77.3}_\boldsymbol{1.0}$    | $\boldsymbol{83.0}_\boldsymbol{0.8}$ | $\boldsymbol{83.6}_\boldsymbol{0.5}$ | $\boldsymbol{79.1}_\boldsymbol{1.6}$ | $\boldsymbol{77.8}_\boldsymbol{2.7}$ | ${76.1}_{0.8}$     | $\boldsymbol{78.9}_\boldsymbol{0.5}$  | $\boldsymbol{79.6}_\boldsymbol{0.3}$    |
>
> **W2**: Thanks for your suggestions. We evaluate our ReFusion on typical knowledge-intensive tasks, such as SQuAD. We also follow the 16-shot setting and evaluate our ReFusion with 5 random seeds. We compare our ReFusion with the baseline method and other SOTA methods [2,3] with the same settings. As shown in Table 2, our ReFusion can also achieve a significant improvement.
>
> | **Dataset** | **T5-Base** | **ReFusion (T5-Base)** | **Splinter [2]** | **FewshotBART [3]** |
> |-------------|-------------|------------------------|------------------|---------------------|
> | SQuAD       | ${46.6}_{3.9}$        | $\boldsymbol{58.6}_\boldsymbol{0.9}$               | ${54.6}$             | ${55.5}_{2.0}$                |
>
> [1] Colin Raffel, Noam Shazeer, Adam Roberts, Katherine Lee, Sharan Narang, Michael Matena, Yanqi Zhou, Wei Li, Peter J. Liu. Exploring the Limits of Transfer Learning with a Unified Text-to-Text Transformer. J. Mach. Learn. Res. 21: 140:1-140:67 (2020).
>
> [2] Ori Ram, Yuval Kirstain, Jonathan Berant, Amir Globerson, Omer Levy. Few-Shot Question Answering by Pretraining Span Selection. ACL/IJCNLP (1) 2021: 3066-3079.
>
> [3] Rakesh Chada, Pradeep Natarajan. FewshotQA: A simple framework for few-shot learning of question answering tasks using pre-trained text-to-text models. EMNLP (1) 2021: 6081-6090.

---

> ### Author Response · Authors · 2023-11-17
>
> **W3**: Thanks for your suggestions. Although we cannot access the source codes of current modern language models, such as GPT-4, which always act as black-box functions, this accessibility issue should not be a major concern in academic research. For a concrete example, although many technical details of Intel CPU/NVIDIA GPU are not available, researchers still work on the field of electronic design automation. The published works still inspire the industry and the real-world practice on chip design [4, 5, 6].
>
> As the main structure of popular large language models follows Transformer, our methodology could be promptly deployed and tested on models like GPT-4. Concretely, our ReFusion can also be built as a library and used with a few lines of code. For example, the following codes demonstrate how to augment the pretrained models with our ReFusion. The code is also updated in the new supplemental.
>
> ```python
> from transformers import AutoTokenizer, AutoModel, AutoConfig
> from refusion import refusion
> tokenizer = AutoTokenizer.from_pretrained("roberta-base")
> config = AutoConfig.from_pretrained("roberta-base")
> config.is_decoder = True
> model = AutoModel.from_pretrained("roberta-base", config=config)
> retriever = refusion.load_retriever(retriever_path, nprobe, topk, device)
> model = refusion.replace_retrieval_module(retrieval_args, retriever, model)
> inputs = "Hello, my dog is cute"
> retriever.retrieve(inputs)
> inputs = tokenizer(inputs, return_tensors="pt")
> outputs = model(**inputs)
> ```
>
> Considering fine-tuning modern language models is a costly task, we evaluate our ReFusion with state-of-the-art parameter-efficient fine-tuning methods, such as LoRA [7]. Table 3 shows that our ReFusion with LoRA can also improve the model performance by 0.9 on average.
>
>
> Table 1. Results with LoRA.
> | **Methods**        | **SST-2** | **SST-5** | **MR**   | **CR**   | **MPQA** | **Subj** | **TREC** | **CoLA** | **Avg-S** |
> |--------------------|-----------|-----------|----------|----------|----------|----------|----------|----------|-----------|
> | RoBERTa-large (LoRA)            | $\boldsymbol{91.4}_\boldsymbol{0.1}$      | $\boldsymbol{45.2}_\boldsymbol{4.3}$      | ${87.5}_{0.3}$     | ${90.2}_{0.9}$     | ${78.9}_{2.3}$     | ${85.3}_{0.7}$     | ${41.4}_{0.3}$     | ${4.0}_{1.8}$     | ${65.5}_{0.0}$      |
> | ReFusion (LoRA) | ${91.1}_{0.7}$  | ${44.8}_{0.8}$  | $\boldsymbol{88.5}_\boldsymbol{0.4}$ | $\boldsymbol{90.8}_\boldsymbol{0.1}$ | $\boldsymbol{81.1}_\boldsymbol{0.6}$ | $\boldsymbol{89.1}_\boldsymbol{0.6}$ | $\boldsymbol{46.5}_\boldsymbol{4.9}$ | ${2.5}_{0.2}$ | $\boldsymbol{66.8}_\boldsymbol{0.4}$  |
>
> Table 1 (cond.). Results with LoRA.
> | **Methods**        | **MNLI** | **MNLI-mm** | **SNLI** | **QNLI** | **RTE**  | **MRPC** | **QQP**  | **Avg-P** | **Avg-all** |
> |--------------------|----------|-------------|----------|----------|----------|----------|----------|-----------|-------------|
> | RoBERTa-large (LoRA)            | $\boldsymbol{57.0}_\boldsymbol{0.1}$      | $\boldsymbol{57.4}_\boldsymbol{0.6}$      | $\boldsymbol{68.9}_\boldsymbol{5.2}$     | ${62.4}_{0.5}$     | $\boldsymbol{56.7}_\boldsymbol{3.0}$     | $\boldsymbol{72.3}_\boldsymbol{6.4}$     | ${59.4}_{1.3}$     | ${62.0}_{0.6}$     | ${63.8}_{0.3}$      |
> | ReFusion (LoRA) | ${56.8}_{0.1}$  | ${57.3}_{0.8}$  | ${67.8}_{0.6}$ | $\boldsymbol{64.2}_\boldsymbol{1.1}$ | ${56.5}_{1.8}$ | ${71.2}_{3.7}$ | $\boldsymbol{62.4}_\boldsymbol{3.7}$ | $\boldsymbol{62.3}_\boldsymbol{1.2}$ | $\boldsymbol{64.7}_\boldsymbol{0.3}$  |
>
> [4] Rong Ye, Ting Wang, Feng Yuan, Rakesh Kumar, Qiang Xu. On reconfiguration-oriented approximate adder design and its application. ICCAD 2013: 48-54. **(Best paper)**
>
> [5] Sawan Singh, Arthur Perais, Alexandra Jimborean, Alberto Ros. Exploring Instruction Fusion Opportunities in General Purpose Processors. MICRO 2022: 199-212.
>
> [6] Björn Gottschall, Lieven Eeckhout, Magnus Jahre. TIP: Time-Proportional Instruction Profiling. MICRO 2021: 15-27. **(Best paper)**
>
> [7] Edward J. Hu, Yelong Shen, Phillip Wallis, Zeyuan Allen-Zhu, Yuanzhi Li, Shean Wang, Lu Wang, Weizhu Chen. LoRA: Low-Rank Adaptation of Large Language Models. ICLR 2022.

---

> ### Author Response · Authors · 2023-11-17
>
> **Q1**: Thanks for your suggestions. We have revised the caption of Figure 2.
>
> **Q2**: Thanks for your suggestions. We have revised the Related Work section and added the descriptions of context-augmentation methods [8, 9, 10] and retrieval fusion methods [11, 12, 13]. Please check page 9 for the new related works.
>
> [8] Ori Ram, Yoav Levine, Itay Dalmedigos, Dor Muhlgay, Amnon Shashua, Kevin Leyton-Brown, Yoav Shoham. In-Context Retrieval-Augmented Language Models. CoRR abs/2302.00083 (2023).
>
> [9] Gautier Izacard, Edouard Grave. Leveraging Passage Retrieval with Generative Models for Open Domain Question Answering. EACL 2021: 874-880.
>
> [10] Zhicheng Guo, Sijie Cheng, Yile Wang, Peng Li, Yang Liu. Prompt-Guided Retrieval Augmentation for Non-Knowledge-Intensive Tasks. ACL (Findings) 2023: 10896-10912.
>
> [11] Patrick S. H. Lewis, Ethan Perez, Aleksandra Piktus, Fabio Petroni, Vladimir Karpukhin, Naman Goyal, Heinrich Küttler, Mike Lewis, Wen-tau Yih, Tim Rocktäschel, Sebastian Riedel, Douwe Kiela. Retrieval-Augmented Generation for Knowledge-Intensive NLP Tasks. NeurIPS 2020.
>
> [12] Sebastian Borgeaud, Arthur Mensch, Jordan Hoffmann, Trevor Cai, Eliza Rutherford, Katie Millican, George van den Driessche, Jean-Baptiste Lespiau, Bogdan Damoc, Aidan Clark, Diego de Las Casas, Aurelia Guy, Jacob Menick, Roman Ring, Tom Hennigan, Saffron Huang, Loren Maggiore, Chris Jones, Albin Cassirer, Andy Brock, Michela Paganini, Geoffrey Irving, Oriol Vinyals, Simon Osindero, Karen Simonyan, Jack W. Rae, Erich Elsen, Laurent Sifre. Improving Language Models by Retrieving from Trillions of Tokens. ICML 2022: 2206-2240.
>
> [13] Zonglin Li, Ruiqi Guo, Sanjiv Kumar. Decoupled Context Processing for Context Augmented Language Modeling. NeurIPS 2022.

---

> ### Author Response · Authors · 2023-11-22
> **Eager for responses**
>
> Dear Reviewer aM36:
>
> Thanks again for all of your constructive suggestions, which helped us improve the quality and clarity of the paper!
>
> Our rebuttal has been posted for a while. We have not heard any post-rebuttal response yet. Please do not hesitate to let us know if there are any additional clarifications that we can provide, as we would like to convince you of the merits of the paper.
>
> We appreciate your responses. Thanks!
>
> Sincerely, Authors

---

### Official Review · Reviewer_9abu · 2023-11-10

**Soundness:** 3 good
**Presentation:** 3 good
**Contribution:** 2 fair
**Rating:** 8
**Confidence:** 4

**Summary:**

With an aim to inject external knowledge into language models (LMs) for performing non-knowledge-intensive (NKI) tasks, including sentiment analysis, opinion polarity analysis, grammatical judgment, natural language inference, etc., towards which retrieval-based augmentation has been struggling as a solution due to the incapability of LMs to handle long sequence during the process, this paper proposes "ReFusion" a computation-efficient retrieval representation fusion framework as a solution to handle the same. Towards this, at first, top-k similar sentences are retrieved using an online retrieval module; thereafter, based on the neural architecture search (NAS) optimal ranking scheme, several modules are replaced with either fusion module with the reranker-based scheme, the fusion module with the ordered-mask-based scheme, or the original module. Authors have conducted comprehensive experiments across 15 NKI tasks comprising 8 single-sentence tasks and 7 sentence-pair tasks. Experimentations are performed using the RoBERTa LMs where ReFusion achieves state-of-the-art performance on 5 tasks over 8 single-sentence tasks and also achieves state-of-the-art performance on 5 tasks over 7 sentence-pair tasks. The study of the averaged performance reflects ReFusion is unbeatable in any of the cases over all 15 NKI tasks. Authors have also performed an ablation study to understand the importance of the sub-components, and the results reported validate that combining different ranking schemes on different tasks is necessary. Moreover, ranking schemes are not always suitable for every layer in LMs. Therefore, ReFusion disables the fusion module at some layers, thus integrating all effective candidate fusion modules.

**Strengths:**

1. The authors claim to be the first to propose fusing the representations of retrievals directly into models to solve the performance and efficiency bottleneck of prompt-based techniques.
2. Ranking-based weighted representations are fused to LMs to enhance their performance over NKI tasks, which helps to overcome the issue of LMs' inability to handle long sequences during the retrieval-based augmentation processes.

**Weaknesses:**

1. The fusion of representation directly to LMs' layers is based on top-k sentence retrieval based on similarity. However, the authors need to study the impact of some more metrics while retrieving top-k similar sentences.
2. This approach is suitable for situations where adaptability is more important than time constraints.

**Questions:**

1. Have authors explored the work in ''Unlimiformer: Long-Range Transformers with Unlimited Length Input (Bertsch et al., 2023)'' where authors claimed that their LM can handle unlimited input sequences?
2. Authors should consider extending this study to other LMs except Roberta.

---

> ### Author Response · Authors · 2023-11-22
>
> **W1**: Thanks for your suggestions. The similarity metric used in our paper is L2 distance. We also build the retriever based on other similarity metrics, such as inner product, which is also a popular similarity metric. We evaluate our ReFusion with inner product metric on the representative tasks used in the ablation study. As shown in Table 1, ReFusion with inner product (IP) achieves a comparable performance compared to that of ReFusion with L2 distance.
>
> Table 1. Results with different similarity metrics.
> | Method        | MPQA | Subj | TREC | SNLI | QNLI | RTE  | Avg |
> |---------------|------|------|------|------|------|------|-----|
> | ReFusion (L2) | ${86.7}_{1.1}$ | ${92.5}_{0.8}$ | ${90.3}_{3.7}$ | ${80.6}_{1.4}$ | ${73.0}_{1.1}$ | ${70.9}_{2.3}$ | ${82.3}_{0.5}$ |
> | ReFusion (IP) | ${86.4}_{0.8}$ | ${92.1}_{2.1}$ | ${91.2}_{2.6}$ | ${80.9}_{1.2}$ | ${73.3}_{1.2}$ | ${71.8}_{3.6}$ | ${82.6}_{0.7}$ |
>
> **W2**: Thanks for your comments. Although our ReFusion reveals great adaptability, the inference efficiency of our ReFusion is still comparable. Table 2 shows the inference latency per sentence of our ReFusion, the baseline LM-BFF, and the retrieval-augmented baseline CA-512.
>
> Compared to CA-512, our ReFusion can significantly reduce the overall inference latency by about 23% on average. The longer retrieving latency is due to more retrievals. Compared to LM-BFF, ReFusion increases the overall inference latency. This is mainly due to the time cost of online retrieving neighbors from the disk. However, this process can be further accelerated by GPUs or prefetch strategies. ReFusion increases by only about 1.94 ms compared to LM-BFF when only considering the model inference. However, the performance of ReFusion is improved by about 2.5 on average compared to LM-BFF. In a word, our ReFusion makes a better trade-off between performance and efficiency.
>
> Table 1. Inference latency per sentence of baselines and our ReFusion. The results are shown in microseconds. The number of retrievals for CA-512 is 16 due to the max length limit. The number of retrievals for ReFusion is 64. The second column indicates the name of the latency breakdown, i.e., "Model" denotes the model inference stage, "Retrieve" denotes the retrieving stage, "Overall" denotes the overall latency. The backbone model of three methods is RoBERTa-large. Lower is better.
> | **Methods**  |      | **SST-2** | **SST-5** | **MR**   | **CR**   | **MPQA** | **Subj** | **TREC** | **CoLA** | **MNLI** | **MNLI-mm** | **SNLI** | **QNLI** | **RTE**  | **MRPC** | **QQP**  | **Avg**|
> |-----------|--------|-----------|-----------|----------|----------|----------|----------|----------|----------|----------|-------------|----------|----------|----------|----------|----------|------|
> | LM-BFF   |   Model | $7.37$ | $7.45$ | $7.40$ | $7.39$ | $7.35$ | $7.41$ | $7.37$ | $7.37$ | $14.85$ | $14.88$ | $14.84$ | $7.53$ | $14.84$ | $7.50$ | $7.47$ | $9.40$ |
> | CA-512     |  Retrieve   | $7.33$ | $7.50$ | $7.61$ | $8.09$ | $5.71$ | $8.46$ | $5.72$ | $6.02$ | $12.62$ | $12.69$ | $10.08$ | $13.60$ | $18.21$ | $11.84$ | $10.93$ | $9.76$ |
> |                  |   Model  | $31.98$ | $31.98$ | $32.06$ | $32.08$ | $31.69$ | $32.24$ | $31.86$ | $31.92$ | $15.48$ | $15.56$ | $15.55$ | $33.25$ | $15.63$ | $33.16$ | $33.07$ | $27.83$ |
> |                  |   Overall  | $39.31$ | $39.48$ | $39.67$ | $40.17$ | $37.40$ | $40.70$ | $37.58$ | $37.94$ | $28.10$ | $28.25$ | $25.63$ | $46.85$ | $33.84$ | $45.00$ | $44.00$ | $37.59$ |
> | ReFusion  |  Retrieve   | $12.96$ | $12.49$ | $12.93$ | $13.60$ | $11.27$ | $13.56$ | $11.43$ | $11.69$ | $22.82$ | $23.06$ | $20.51$ | $23.85$ | $28.79$ | $21.34$ | $21.12$ | $17.43$ |
> |                 |  Model   | $9.08$ | $9.09$ | $8.98$ | $9.04$ | $8.95$ | $9.03$ | $9.02$ | $8.99$ | $17.16$ | $17.18$ | $17.15$ | $9.67$ | $17.45$ | $9.72$ | $9.61$ | $11.34$ |
> |                 |  Overall   | $22.04$ | $21.58$ | $21.91$ | $22.64$ | $20.22$ | $22.59$ | $20.45$ | $20.68$ | $39.98$ | $40.24$ | $37.66$ | $33.52$ | $46.24$ | $31.06$ | $30.73$ | $28.77$ |

---

> ### Author Response · Authors · 2023-11-22
>
> **Q1**: Thanks for your comments. There are several key differences between our ReFusion and Unlimiformer. Firstly, our ReFusion can be applied to any transformer-based model while Unlimiformer is only designed for encoder-decoder models. Secondly, the retrieval module of Unlimiformer only resides in memory while our ReFusion has an on-disk retrieval module, which is not limited by the memory capacity and has a larger knowledge database. Thirdly, Unlimiformer uses a cross-attention module to fuse retrievals while our ReFusion uses a more computation-efficient module. Fourthly, Unlimiformer prefers to retrieve from long-range inputs, while our ReFusion prefers to retrieve from external knowledge databases. Finally, the goal of Unlimiformer, making transformers process unlimited length input, is different from that of our ReFusion, efficiently introducing retrievals. Actually, our ReFusion is orthogonal to Unlimiformer.
>
> **Q2**: Thanks for your suggestions. We applied our ReFusion to the popular autoregressive language models, such as T5 [1]. Please refer to Table 1 in the response to the **Reviewer aM36**.  Experimental results show that our ReFusion can be adapted to autoregressive models, improving the model performance by 1.4 on average.
>
> [1] Colin Raffel, Noam Shazeer, Adam Roberts, Katherine Lee, Sharan Narang, Michael Matena, Yanqi Zhou, Wei Li, Peter J. Liu. Exploring the Limits of Transfer Learning with a Unified Text-to-Text Transformer. J. Mach. Learn. Res. 21: 140:1-140:67 (2020)

---

### Meta-Review · Area_Chair_WtG5 · 2023-12-11

**Metareview:**

The paper introduces a new retrieval augmented model. The rather than concatenating the retrieved context to the model input, the paper instead proposes fusing representations of the retrieved context directly into the model's hidden states. There is some disagreement amongst reviewers. While most reviews are positive, and appreciate the extensive evaluation on many tasks, Reviewer aM36 raises concerns about the scope of the paper: it is applied only to encoder-only models with short context (Roberta) on non-knowledge intensive tasks. While I agree that these are valid points that will limit the impact of the paper, I think that given the continued popularity of Roberta-like models in resource constrained settings, overall the paper makes a useful contribution and can be accepted to ICLR.

**Justification For Why Not Higher Score:**

The scope of the paper is quite limited

**Justification For Why Not Lower Score:**

The main weakness is limited scope, but I think some people might find this method useful

---

### Decision · Program_Chairs · 2024-01-16

Accept (poster)